# Eight-Day Fast and a Single Bout of Exercise: The Effect on Serum Methylarginines and Amino Acids in Men

**DOI:** 10.3390/nu15132981

**Published:** 2023-06-30

**Authors:** Joanna Reczkowicz, Jakub Kortas, Ulana Juhas, Malgorzata Zychowska, Andzelika Borkowska, Karol Pilis, Ewa Ziemann, Zuzanna Sobol, Jedrzej Antosiewicz

**Affiliations:** 1Department of Bioenergetics and Physiology of Exercise, Medical University of Gdansk, 80-210 Gdansk, Poland; joanna.reczkowicz@gumed.edu.pl (J.R.); ulana.juhas@gumed.edu.pl (U.J.); andzelika.borkowska@gumed.edu.pl (A.B.); 2Department of Health and Life Sciences, Gdansk University of Physical Education and Sport, 80-336 Gdansk, Poland; jakub.kortas@awf.gda.pl; 3Department of Biological Foundation of Physical Culture, Faculty of Health Science and Physical Culture, Kazimierz Wielki University, 85-064 Bydgoszcz, Poland; malgorzata.zychowska@ukw.edu.pl; 4Department of Health Sciences, Jan Długosz University in Czestochowa, 42-200 Czestochowa, Poland; k.pilis@ujd.edu.pl; 5Department of Athletics, Strength and Conditioning, Poznan University of Physical Education, 61-871 Poznan, Poland; ziemann.ewa@gmail.com; 6Masdiag Sp. z o.o., 33 Stefana Żeromskiego St., 01-882 Warsaw, Poland; zuzanna.sobol@masdiag.pl

**Keywords:** methylarginines, amino acids, dimethylamine, fasting

## Abstract

Changes in serum concentration of methylarginines and amino acids after exercise are well documented, whereas the effects of exercise applied together with fasting are still debated and not thoroughly studied. Thus, we hypothesised that alterations in methylarginines such as ADMA, SDMA and L-NMMA might be responsible for decreased exercise performance after 8 days of fasting. Additionally, we propose that conditions in which the human body is exposed to prolonged fasting for more than a week elicit a distinctly different response to exercise than after overnight fasting. A group of 10 healthy men with previous fasting experience participated in the study. The exercise test was performed until exhaustion with a gradually increasing intensity before and after the 8-day fast. Blood samples were collected before and immediately after exercise. ADMA, SDMA, L-NMMA, dimethylamine and amino acids were analysed in serum samples by ID-LC-MS/MS. SDMA, L-NMMA and dimethylamine significantly decreased after 8 days of fasting, whereas ADMA did not change. BCAA, Phe, alanine and some other amino acids increased after fasting. Exercise-induced changes in amino acids were distinct after an 8-day fast compared to overnight fasting. A decrease in physical performance accompanied all of these alterations. In conclusion, our data indicate that neither methyl-arginine changes nor the Trp/BCAA ratio can explain exercise-induced fatigue after fasting. However, the observed decrease in hArg concentration suggests the limited synthesis of creatine, possibly contributing to reduced physical performance.

## 1. Introduction

People often apply fasting to increase their health and general well-being and to promote longevity [1]. While prolonged starvation induces autophagy, it also increases protein degradation, enhances lipids metabolism and reduces carbohydrate metabolism [2]. Some of these changes are desirable. However, some metabolic changes can be deleterious [3]. For example, high protein degradation decreases skeletal muscle mass and impairs mitochondria function, negatively influencing overall mobility [3]. Conversely, an impairment in carbohydrate metabolism can lead to a decrease in physical performance. Limited availability of carbohydrates during starvation can also change fatty acid metabolism as acetyl-CoA entry to the Krebs cycle is limited by oxaloacetate derived from a glucose metabolite pyruvate. Metabolic adaptive mechanisms in the body that permit survival for prolonged periods include the provision of glucose from liver glycogen in the first 18–24 h and hepatic gluconeogenesis derived mainly from muscle amino acids and glycerate after that [4]. Fasting also promotes lipolysis to release non-esterified fatty acids, resulting in converting into acetyl-CoA and then to ketone bodies in the liver. Ketone bodies provide a metabolic fuel for the brain, skeletal muscle and other tissues [5]. From the energy perspective, lipolysis and ketone bodies production upregulation enable the brain to shift from glucose utilisation to ketone bodies. This mechanism helps preserve muscle protein and allows human beings to withstand extended periods of fasting, lasting several days or even weeks [6].

Physiological protein turnover during fasting can be associated with increased serum levels of amino acids. Furthermore, proteolysis of proteins containing methylated arginine residues releases various N-methylarginine (L-NMMA) derivatives into the blood and explicitly into plasma. Protein substrates of arginine residues can undergo posttranslational modification as a methylation reaction catalysed by a family of enzymes known as protein arginine methyltransferases [7]. Therefore, when these proteins undergo proteolytical degradation, besides nonmodified amino acids additionally L-NMMA, symmetric dimethylarginine (SDMA), asymmetric dimethylarginine (ADMA), dimethylamine can be released into the bloodstream [8]. The physiological function of these metabolites has yet to be fully documented; however, their deleterious action has been demonstrated [9]. Of these three L-NMMA residues produced in mammals, L-NMMA, ADMA and indirectly SDMA inhibit nitric oxide synthase (NOS). Thus, it has been proposed that endogenously produced L-NMMA, SDMA and ADMA, when accumulated in disease conditions, may contribute to pathology by inducing vasoconstriction and limiting blood flow [9]. Fasting is widely recognised to be correlated with reduced exercise performance, which can be attributed to the limited availability of carbohydrates and impairment of energetic metabolism. However, the exact mechanism of fasting-inducing fatigue remains unclear. Moreover, some studies have shown that performance is impaired, whereas others have shown no effect [10]. Here, we hypothesised that adverse accumulation of methylarginines may contribute to lower performance. Additionally, alterations in blood amino acids could also supply exercise-induced fatigue. It is anticipated that chain amino acids will decrease during exercise, leading to increased tryptophan transport into the central nervous system and subsequent serotonin synthesis, which can contribute to central fatigue [11]. Therefore, the primary objective of the present study was to investigate whether an eight-day water fasting period in a group of healthy men affects the serum concentration of methylarginines and amino acids and whether these changes are related to exercise performance.

## 2. Materials and Methods

### 2.1. Participant’s Inclusion and Exclusion Criteria

The study assumed specific inclusion and exclusion criteria. To qualify for the study, participants had to have previous experience fasting for more than three days, be between the ages of 30 and 70 years, have a body weight between 60 and 100 kg, and have a body mass index (BMI) within the range of 20–29.9 kg/m^2^. They were also required to have no chronic diseases and to maintain systolic blood pressure within the range of 100–140 mm Hg and diastolic blood pressure within the range of 60–90 mm Hg. On the other hand, individuals who smoked cigarettes, used medications, had any diseases or used potent stimulants or psychoactive substances were excluded from the study. Failure to complete the test procedure was also the criterion for exclusion.

The research design did not include a separate control group, as the analysis examined intervention effects based on participants’ data.

### 2.2. Group Characteristics

The study included 14 healthy men who had previously undergone a range of fasting periods, with the last being at least six months before the study (age: 54.40 ± 13.16 years old).

All eligible participants underwent a medical examination, which found no contraindications for exercise. The participants reported engaging in moderate-intensity physical activity in the form of yoga. The basic somatic parameters of the participants are shown in Table 1. One participant was excluded from the analysis due to vitamin D supplementation (approximately 10,000 IU per day) for several weeks before the study. During the fasting period, the participants consumed only mineral water, which contained an average amount of ions, and were not allowed to consume any food or drink containing calories. No adverse effects of fasting were recorded during the study.

### 2.3. Exercise Test

An exercise test was conducted using an Excalibur Sport cycle ergometer (Lode B.V., Groningen, The Netherlands), where the workload was progressively increased until the point of exhaustion was reached by the participants. The initial load was set at 60 W and was increased every 3 min by 30 W. The test was terminated when the subject’s oxygen uptake stabilised at the maximum level or decreased, when the subject could not maintain the pedalling rhythm, or when the heart rate failed to increase, stabilised at its maximum level or started to decrease. This exercise test was conducted before and after eight days of complete fasting.

Individual participants’ overall physical fitness levels were not assessed as the study aimed to compare the effects of exercise before and after fasting within the same subjects.

### 2.4. Blood Collection

During the study, biochemical analyses were performed on study participants. Medical professionals collected 10 millilitres of venous blood from each participant at four time points, i.e., before and after the intervention at baseline and 1 h after a single bout of exercise on a cycle ergometer. Blood samples were stored correctly and preserved at −80° for further analysis.

### 2.5. Samples Analysis

Chemicals: The commercially available unlabelled and isotope-labelled amino acids standards (ISTD) were purchased from Cambridge Isotope Laboratories (Tewksbury, MA, USA), Sigma Aldrich (Schnelldorf, Germany) and Toronto Research Chemicals (Toronto, ON, Canada). LC-MS grade solvents and additives (methanol, water, ammonium formate, formic acid) were purchased from VWR (Darmstadt, Germany). AccQ-Tag Ultra Derivatization Kit was obtained from Waters (Milford, MA, USA), 3.0 N HCl in n-butanol from Sigma Aldrich.

Sample preparation: Isotope dilution mass spectrometry (ID-LC-MS/MS) was used to analyse the concentration of amino acids in plasma and serum samples. Analytes were divided into two groups with separate sample preparation and analysis protocols. Before analysis samples were collected, frozen and stored at −20 °C. The LC-MS/MS analysis was performed on a Shimadzu HPLC system with an autosampler HTC PAL (Zwingen, Switzerland) coupled to a 4000API (Sciex, Framingham, MA, USA). Quantitative results were obtained using solvent-based calibration curves.

#### 2.5.1. Amino Acids Analysis

Amino acids (Ala, Arg, Asn, Asp, Gln, Glu, Gly, His, Ile, Leu, Lys, Met, Phe, Pro, Ser, Thr, Trp, Tyr, Val, Cit, hArg, Orn, Sarc, bAla, GABA, Tau) were analysed using ID-LC-MS/MS technique preceded by sample preparation based on double protein precipitation and derivatisation by n-butylation. The serum or plasma sample (10 µL) was precipitated in a polypropylene 96-well plate with 50 µL of water: methanol 20:80 ISTD solution and 100 µL of 80% methanol in water with the addition of 0.2 M hydrogen chloride (15 min, 450 RPM, RT). The supernatant was obtained by centrifugation (10 min, 3000 RPM, RT) and transferred (50 µL) to a new polypropylene 96-well plate for additional precipitation. That step was performed by adding 100 µL of acetonitrile (15 min, 450 RPM, RT) and centrifugation (10 min, 3000 RPM, RT). After this time, 20 μL of supernatant was collected from each well and transferred to a polypropylene 96-well plate, which was dried at 50 °C under a stream of nitrogen for about 12 min. Subsequently, 25 μL of derivatisation agent (3.0 N HCl in n-butanol) was added to each well. The mixture was kept at 60 °C for 25 min. After that, the 96-well plate was dried at 50 °C under a stream of nitrogen for about 12 min. The residue was dissolved in a mixture of methanol/water (5:95) with the addition of 0.1% formic acid. A 5 μL aliquot was subjected to LC-MS/MS analysis.

The chromatographic analyses were performed using Agilent (Santa Clara, CA, USA) Zorbax Eclipse XDB-C18 1.8 µm (50 × 4.6 mm) column at a flow rate of 0.8 mL/min (40 °C) in the linear gradient of water (A) and methanol: acetonitrile (1:1) (B), both with the addition of 0.1% formic acid.

Quantitative results were obtained using solvent-based calibration curves with different dynamic ranges depending on their physiological concentrations: 1.25–1625 ηmol/mL for Ala, Asn, Asp, Gln, Glu, Gly, His, Ile, Leu, Lys, Met, Phe, Pro, Ser, Thr, Trp, Tyr, Val, Cit, Orn and Tau; 0.0625–1625 nmol/mL for Arg; 0.125–1625 nmol/mL for hArg and Sarc; 0.25–1625 nnom/mL for bAla and GABA.

#### 2.5.2. Arginine-Related Amino Acids and Metabolites

Amino acids metabolites (dimethylamine, NG, NG-dimethylarginine (ADMA), NG, NG’-dimethyl-L-arginine (SDMA), NG-monomethyl-L-arginine (methylarginine)) were analysed using an ID-LC-MS/MS technique preceded by sample preparation based on protein precipitation and specific derivatisation using a commercially available AccQ-Tag reagent (Waters). The serum or plasma sample (50 µL) was precipitated in a deep-well 96-well plate with 200 µL of water: methanol 20:80 ISTD solution (15 min, 1100 RPM, RT). The supernatant was obtained by centrifugation (10 min, 3000 RPM, RT) and transferred (10 µL) to a new polypropylene 96-well plate for the derivatisation procedure. The derivatisation procedure was conducted according to the AccQ-Tag Ultra kit manual. In total, 10 µL of supernatant, 75 µL of Borate Buffer (AccQ-Tag Ultra) and 15 µL of reagent 2A (AccQ-Tag Ultra) were mixed (5 s, 450 RPM, RT), equilibrated (1 min) and exposed to elevated temperature (55 °C, 10 min). After this, the derivatisation sample was diluted with the addition of 100 µL water and injected directly into the LC-MS/MS system. A 5 μL aliquot was subjected to LC-MS/MS analysis.

The chromatographic analyses were performed using Agilent (Santa Clara, CA, USA) Zorbax Eclipse XDB-C18 1.8 µm (50 × 4.6 mm) column at a flow rate of 0.8 mL/min (40 °C) in the linear gradient of water (A) and acetonitrile (B), both with the addition of 10 mM ammonium formate. Quantitative results were obtained using solvent-based calibration curves with different dynamic ranges depending on their physiological concentrations: 0.01–10 µg/mL for ADMA, 0.01–10 µg/mL for SDMA, 0.01–10 µg/mL for methylarginine and 0.01–10 µg/mL for Dimethylamine.

### 2.6. Ethics

The study included participants who had prior experience with fasting, and the study protocol was designed to align with their ongoing fasting routine. Throughout the study, the participants were closely monitored by medical staff who provided information regarding the potential adverse health consequences of fasting. As a precautionary measure, the male participants were placed under medical supervision for three days before the start of the study, throughout the eight-day fasting period, and for three days following its conclusion. Informed consent was obtained from all participants, and they were fully informed about the study’s purpose and procedures before providing written consent. The study was approved by the Committee for Ethics in Scientific Research of Jan Dlugosz University in Czestochowa (Poland; KE-0/1/2019; 5 March 2019) and was conducted following the ethical principles outlined in the Declaration of Helsinki for medical research involving human subjects.

### 2.7. Statistical Analysis

Statistical analysis was conducted using Statistica 13.1 software (StatSoft, Tulsa, OK, USA). All values are reported as the mean ± standard deviation (SD). Furthermore, the 95% confidence interval for changes and the d Cohen effect size were computed. The homogeneity of dispersion from a normal distribution was assessed using the Shapiro–Wilk test. The homogeneity of variance was evaluated using the Brown–Forsythe test. Initially, the changes following an 8-day fasting period were examined. In cases where the results demonstrated homogeneity, a paired *t*-test analysis was employed to determine significant differences. For cases with heterogeneous results, the Wilcoxon signed-rank test was used. Subsequently, the response to a single bout of exercise was analysed by comparing changes before and after the intervention. An analysis of variance (ANOVA) for repeated measures was performed in instances of homogeneity. For cases of heterogeneity, the ANOVA Friedman’s test was employed. The level of significance was set at *p* < 0.05.

## 3. Results

The final analysis considered complete samples from 10 participants. Generally, in response to 8 days of fasting, a statistically significant increase in the beta-hydroxybutyrate level was observed (0.28 ± 0.17 vs. 4.28 ± 1 mmol/L, *p* = 0.01). Moreover, the decrease in response to a single bout of exercise was noticed both before (Δ = −0.11, *p* = 0.01) and after (Δ = −0.70, *p* = 0.01) 8-day fasting, but after the intervention, the decrease was statistically significantly more extensive (*p* < 0.01).

The results of changes in anthropometric measurements and physical performance are summarised in Table 1, illustrating the effects induced by fasting for more than a week. The intervention resulted in significant changes in body composition, including a reduction in body weight, body fat, fat-free mass, total body water and BMI. In addition, oxygen capacity indicating physical activity status dropped statistically.

The results presented in Table 2 refer to the blood serum analytes before and after the 8-day fasting, where all L-NMMA concentrations decreased, and the changes in SDMA, dimethylamine, and L-NMMA changed statistically significantly. As for amino acids, there was a significant increase in BCAAs (leucine, isoleucine and valine), alanine, beta-alanine, glycine, lysine, methionine, phenylalanine, proline and tyrosine. The concentration of citrulline, hArg, and tryptophan to BCAA ratio relevantly decreased after the intervention.

A decrease in amino acids and L-NMMA concentrations following a single bout of exercise compared to pre- and post-intervention levels was observed. In contrast to the post-intervention effect, BCAAs decreased in this case, and this change was statistically significant. Similar changes were observed in alanine, methionine and glycine concentrations. On the other hand, changes in concentrations of glutamic acid, ornithine, phenylalanine and taurine were observed after a single bout of exercise. Still, not after an 8-day fast, these changes were insignificant (Table 3).

It should be noted that methylarginines and tryptophan to BCAA ratio mainly increased, although none of these changes reached statistical significance.

## 4. Discussion

The main finding of this study is that despite accelerated proteolysis, as evidenced by increased concentrations of several amino acids in the serum, there was no adverse accumulation of L-NMMA and ADMA after eight days of fasting. On the contrary, a significant decrease in SDMA and L-NMMA was observed. As mentioned earlier, the increase in protein degradation typically observed during fasting can result in the release of methylated derivatives of arginine, some of which have adverse effects, mainly by inhibiting NO synthesis [9]. However, in this study, fasting did not lead to the expected accumulation of these methylated arginine derivatives. L-NMMA and ADMA are extensively metabolised intracellularly to citrulline and dimethylamine (DMA) in a reaction catalysed by dimethylarginine dimethylaminohydrolase (DDAH), and around 20% of ADMA is excreted by the kidneys [12]. Conversely, SDMA is eliminated almost entirely by renal excretion. Thus, the serum concentration of L-NMMA and ADMA is a net result of its formation and elimination, mainly in the reaction catalysed by the DDAH [13]. The observed decrease in DMA and L-NMMA serum concentration suggests that their urinary excretion is effective during fasting. Thus, contrary to our expectation, starvation-induced proteolysis did not lead to an increase in the concentration of methylated arginine metabolites, which have the potential to inhibit NO synthesis. Furthermore, the observed decrease in SDMA and L-NMMA, along with no change in ADMA and arginine to ADMA ratio, suggests that fasting-induced changes support NO synthesis. Indeed, SDMA, unlike ADMA, does not directly inhibit reactions catalysed by NOS but acts as a competitive inhibitor of L-arginine transport [8]. Therefore, a decrease in SDMA should also be recognised as a change favouring NO synthesis. Despite these favourable changes, a significant decrease in VO_2_ max has been observed. The primary determinant of VO_2_ max is the capacity of the circulatory system to supply an adequate amount to the working skeletal muscles. The changes in methylated arginine derivatives observed after fasting are expected to support nitric oxide synthesis during exercise. Conversely, a significant decrease in skeletal muscle mass could contribute to lower performance. Additionally, the concentration of homoarginine (hArg) decreases significantly after fasting. hArg is a product of a reaction catalysed by L-arginine: glycine amidinotransferase, an enzyme also involved in creatine synthesis. When enzymes use Arg and lysine as substrates, hArg is formed. However, when glycine replaces lysine, guanidino acetic acid is produced, subsequently converted into creatine. Therefore, a low level of hArg suggests that the formation of creatine can also be limited [14]. Since phosphocreatine is crucial for skeletal muscle energetics and intracellular trafficking of high-energy bonds, a decrease in its levels can potentially impair performance [15]. Furthermore, hArg is known to be a less efficient substrate for NOS. However, studies have demonstrated that hArg can increase arginine levels by inhibiting the arginase activity [16,17]. These data suggest that the observed decrease in hArg may impair exercise performance. Another factor that can influence fatigue is TRP to branch chain amino acids ratio. TRP is a substrate for serotonin synthesis in the central nervous system (CNS). The formation of serotonin is limited by TRP availability, and an increase in its formation was claimed to cause fatigue [18]. For example, limiting TRP consumption by animals increases their exercise performance [19]. TRP is transported into CNS by a large neutral amino acids transporter, which also transports BCAA. During exercise, BCAA undergoes extensive metabolism, decreasing serum concentration, while changes in TRP levels during exercise are minimal, leading to an increase in the ratio of TRP to BCAAs. This increase in the TRP/BCAA ratio promotes the transport of TRP into the brain and subsequent serotonin synthesis, which can contribute to fatigue [11,20]. Here, we observed that the ratio TRP/BCAA significantly decreased after fasting at rest. Conversely, the ratio increased after the exercise, although it remained lower than the ratio observed after the exercise performed at baseline. Therefore, based on our data, it does not appear that changes in TRP/BCAA ratio can be responsible for exercise-induced fatigue after fasting.

Interestingly, a single exercise neither before nor after fasting affects methylarginine derivatives, although significant changes in arginine concentration and other amino acids were observed. The effects of short and long (more than three days) fasting on serum amino acid concentration have been studied for several decades. However, limited data regarding the effects of exercise performed before and after fasting are available. Here, we observed that the serum BCAA significantly increased after fasting, consistent with a previously published study [21]. Considering the limited metabolism of aromatic amino acids, an increase in their concentration after fasting suggests a shift in the balance from protein synthesis to protein synthesis to protein degradation. A similar observation has been reported on eight men after 72 h of fasting [22].

## 5. Conclusions

In conclusion, our data indicate that a decrease in physical performance after eight days of fasting cannot be attributed to changes in methylated arginine derivatives or to changes in serum amino acids. The observed decrease in hArg concentration suggests a limited synthesis of creatine, which may reduce overall physical performance. These findings provide valuable insights into the mechanisms related to prolonged starvation. It is essential to underline that the study was performed on healthy men without any medical conditions and with good physical fitness levels.

## Figures and Tables

**Table 1 nutrients-15-02981-t001:** Anthropometric and physiological characteristics of the study participants.

Variable	I	II	Δ	CI Lower	CI Upper	Cohen’s d	t	*p*
BW [kg]	83.52 ± 9.2	77.8 ± 9.4	−5.72	−6.86	−4.58	−3.58	−11.31	**<0.001**
Fat [%]	20.8 ± 3.5	19.71 ± 3.93	−1.09	−1.66	−0.52	−1.38	−4.36	**0.00**
Fat [kg]	17.61 ± 4.49	15.58 ± 4.54	−2.03	−2.45	−1.61	−3.47	−10.97	**<0.001**
FFM [%]	79.19 ± 3.51	80.01 ± 3.59	0.82	0.05	1.58	0.77	2.42	**0.04**
FFM [kg]	65.91 ± 5.36	62.04 ± 5.96	−3.87	−4.96	−2.78	−2.55	−8.07	**<0.001**
TBW [%]	57.99 ± 2.58	58.79 ± 2.91	0.80	0.36	1.25	1.29	4.07	**0.00**
TBW [kg]	48.26 ± 3.91	45.55 ± 4.08	−2.71	−3.48	−1.94	−2.51	−7.95	**<0.001**
BMI [kg/m^2^]	26.16 ± 2.86	24.36 ± 2.85	−1.80	−2.17	−1.43	−3.52	−11.13	**<0.001**
VO_2_ max [mL/min/kg]	44 ± 11.69	37.7 ± 6.57	−6.30	−11.31	−1.29	−0.90	−2.84	**0.02**

Values are means ± SD. I, after overnight fasting; II, after 8 days of fasting; BW, body weight; Fat, body fat; FFM, fat-free mass; TBW, total body water; BMI, body mass index; VO_2_ max, maximal oxygen consumption. Significant changes are bold, *p* < 0.05.

**Table 2 nutrients-15-02981-t002:** Changes in amino acid profile after the 8-day intervention.

	I—Before			II—After			Δ	95% CI		Cohen’s d	*p*
Variable	M	±	SD	M	±	SD		Lower	Upper		
Ala [nmol/mL]	374.21	±	69.60	447.87	±	79.39	73.66	23.84	123.49	1.06	0.01
Arg [nmol/mL]	154.36	±	17.41	155.76	±	19.20	1.40	−9.52	12.31	0.09	0.78
Asn [nmol/mL]	44.50	±	5.20	48.84	±	7.00	4.34	−1.45	10.12	0.54	0.12
Asp [nmol/mL]	36.73	±	5.72	35.69	±	6.85	−1.04	−6.40	4.32	−0.14	0.67
bAla [nmol/mL]	2.60	±	0.56	3.16	±	0.74	0.56	0.19	0.93	1.07	0.01
Cit [nmol/mL]	29.39	±	7.25	21.91	±	5.00	−7.48	−10.93	−4.02	−1.55	<0.01
GABA [nmol/mL]	0.23	±	0.03	0.26	±	0.07	0.03	−0.02	0.07	0.45	0.19
Gln [nmol/mL]	545.68	±	81.75	599.65	±	88.27	53.98	−5.55	113.50	0.65	0.07
Glu [nmol/mL]	169.21	±	26.87	164.95	±	22.19	−4.25	−22.09	13.59	−0.17	0.60
Gly [nmol/mL]	294.53	±	31.73	427.01	±	67.64	132.48	76.03	188.93	1.68	<0.01
hArg [nmol/mL]	1.72	±	0.23	0.83	±	0.29	−0.90	−1.15	−0.65	−2.57	<0.01
His [nmol/mL]	97.15	±	6.81	92.62	±	9.11	−4.53	−10.34	1.28	−0.56	0.11
Ile [nmol/mL]	76.12	±	14.06	156.10	±	34.14	79.98	53.00	106.95	2.12	<0.01
Leu [nmol/mL]	149.17	±	21.91	256.65	±	39.11	107.48	71.18	143.78	2.12	<0.01
Lys [nmol/mL]	161.26	±	33.84	187.51	±	28.49	26.24	2.73	49.76	0.80	0.03
Met [nmol/mL]	26.52	±	4.39	38.38	±	4.81	11.86	8.00	15.72	2.20	<0.01
Orn [nmol/mL]	51.67	±	9.13	55.31	±	9.22	3.64	−0.66	7.94	0.61	0.09
Phe [nmol/mL]	81.28	±	12.18	92.51	±	8.05	11.23	3.91	18.56	1.10	0.01
Pro [nmol/mL]	192.39	±	36.88	239.01	±	53.54	46.62	25.60	67.64	1.59	<0.01
Sarc [nmol/mL]	1.40	±	0.48	1.37	±	0.41	−0.03	−0.28	0.22	−0.09	0.78
Ser [nmol/mL]	135.85	±	20.99	139.65	±	14.62	3.80	−12.79	20.39	0.16	0.62
Tau [nmol/mL]	50.30	±	22.24	57.34	±	21.43	7.05	−9.98	24.07	0.30	0.37
Thr [nmol/mL]	109.49	±	20.46	125.15	±	18.36	15.66	−2.20	33.51	0.63	0.08
Trp [nmol/mL]	48.36	±	10.86	46.44	±	8.57	−1.92	−11.04	7.21	−0.15	0.65
Tyr [nmol/mL]	58.96	±	15.31	72.65	±	10.46	13.70	4.48	22.91	1.06	0.01
Val [nmol/mL]	240.07	±	27.13	409.01	±	66.66	168.94	120.51	217.36	2.50	<0.01
L-NMMA [µmol/L]	0.23	±	0.01	0.18	±	0.02	−0.05	−0.07	−0.03	−1.87	<0.01
Dimethylamine [µmol/L]	3.22	±	1.55	2.13	±	0.41	−1.09	−2.12	−0.07	−0.76	<0.01
SDMA [µmol/L]	2.29	±	0.47	1.89	±	0.49	−0.40	−0.75	−0.04	−0.80	0.03
ADMA [µmol/L]	0.67	±	0.13	0.62	±	0.07	−0.04	−0.13	0.04	−0.37	0.28
Arg/ADMA	238.65	±	53.64	251.29	±	32.11	12.64	−21.09	46.37	0.27	0.42
Trp/BCAA	0.11	±	0.02	0.06	±	0.01	−0.05	−0.07	−0.03	−1.55	<0.01

Values are means ± SD. I, after overnight fasting; II, after 8 days of fasting.

**Table 3 nutrients-15-02981-t003:** Response of a single bout of exercise before and after the intervention.

Variable	I—Change after First Single Bout			II—Change after Last Single Bout			Δ	CI Lower	CI Upper	ANOVA
Ala [nmol/mL]	129.62	±	32.34	−37.98	±	112.06	167.60	89.18	246.02	<0.01
Arg [nmol/mL]	6.54	±	23.00	−21.59	±	20.18	28.13	6.20	50.06	0.01
Asn [nmol/mL]	−0.74	±	5.70	−5.98	±	7.45	5.24	−1.32	11.80	0.11
Asp [nmol/mL]	1.07	±	3.84	−4.40	±	5.50	5.47	0.80	10.14	0.02
bAla [nmol/mL]	0.42	±	0.43	−0.06	±	0.62	0.48	−0.05	1.00	0.07
Cit [nmol/mL]	0.15	±	2.10	−1.51	±	2.62	1.66	−0.69	4.02	0.15
GABA [nmol/mL]	0.06	±	0.05	0.06	±	0.23	0.00	−0.17	0.16	0.13
Gln [nmol/mL]	71.18	±	76.01	−42.83	±	91.25	114.01	30.53	197.50	0.01
Glu [nmol/mL]	14.03	±	26.32	−24.40	±	24.35	38.43	12.81	64.05	<0.01
Gly [nmol/mL]	34.11	±	33.95	−76.26	±	93.22	110.36	43.29	177.44	<0.01
hArg [nmol/mL]	0.11	±	0.13	−0.05	±	0.13	0.17	0.04	0.30	0.02
His [nmol/mL]	6.51	±	6.39	−7.53	±	12.31	14.04	4.55	23.54	0.01
Ile [nmol/mL]	4.23	±	6.18	−30.84	±	19.08	35.07	21.55	48.59	<0.01
Leu [nmol/mL]	6.34	±	10.53	−53.95	±	23.48	60.29	42.77	77.81	<0.01
Lys [nmol/mL]	15.35	±	13.84	−14.92	±	23.88	30.27	11.26	49.27	<0.01
Met [nmol/mL]	2.42	±	1.42	−2.47	±	5.64	4.88	0.99	8.78	<0.01
Orn [nmol/mL]	0.91	±	6.39	−12.94	±	6.23	13.84	7.49	20.20	<0.01
Phe [nmol/mL]	9.22	±	6.22	−12.85	±	14.35	22.07	11.43	32.70	<0.01
Pro [nmol/mL]	5.38	±	14.98	−6.40	±	38.16	11.78	−16.00	39.56	0.38
Sarc [nmol/mL]	0.01	±	0.18	−0.10	±	0.30	0.12	−0.13	0.36	0.32
Ser [nmol/mL]	1.84	±	14.16	−19.13	±	22.04	20.97	2.83	39.10	0.03
Tau [nmol/mL]	4.30	±	10.49	−21.98	±	18.54	26.28	11.63	40.93	<0.01
Thr [nmol/mL]	4.18	±	9.01	−5.80	±	22.37	9.98	−6.38	26.33	0.46
Trp [nmol/mL]	4.54	±	7.23	−4.99	±	9.64	9.53	1.12	17.95	0.03
Tyr [nmol/mL]	5.27	±	4.07	−3.30	±	10.76	8.57	0.78	16.36	0.04
Val [nmol/mL]	12.58	±	19.15	−62.46	±	44.98	75.04	41.82	108.26	<0.01
ADMA [µmol/L]	−0.00	±	0.12	−0.06	±	0.08	0.06	−0.05	0.17	0.29
L-NMMA [µmol/L]	0.00	±	0.02	−0.01	±	0.02	0.02	0.00	0.04	0.11
SDMA [µmol/L]	−0.02	±	0.44	−0.05	±	0.24	0.04	−0.36	0.43	0.23
Dimethylamine [µmol/L]	0.17	±	0.73	0.07	±	0.50	0.11	−0.58	0.79	0.23
Arg/ADMA	−35.71	±	375.48	135.78	±	141.08	−171.49	517.74	−161.44	0.31
Trp/BCAA	0.01	±	0.02	0.01	±	0.01	0.00	−0.01	0.01	0.70

Values are means changes after exercise ± SD. I, after overnight fasting; II, after 8 days of fasting.

## Data Availability

Datasets analysed during the current study will be available at the end of the project they are part of (Grant from the National Science Centre, Poland—number 2020/37/B/NZ7/01794).

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
