# Peer review of "Eight-Day Fast and a Single Bout of Exercise: The Effect on Serum Methylarginines and Amino Acids in Men"

_nutrients, 2023, doi:10.3390/nu15132981_

Round 1
Reviewer 1 Report
The main issue of the paper is that I cannot assess the statistical analysis as it is not described in the text. Moreover, I found the discussion as an extension of the results while this section must the part to move forward the research field. How does skeletal muscle metabolism respond to fasting? This would help to discuss your data and/ or introduce the researach topic some interesing data here (PMID: 36302454).
Why was not measured NO? The authors extensively discuss how their results would influence NO biology.
I think that the manuscript need some lenguage editing some parts are hard to follow
Moderate comments:
Reference for L45-47.
References for L57-60
References forL65
Reference 6 must be further explained. Why inhibition of NOS is a pathological condition, explain.
Explain reference 7 in more detail. Why is unclear?
L 78- Peripheral or central fatigue?
Was the blood collected immediately after exercise?
L217. “As for oxygen capacity indicating physical activity status, it decreased although without statistical significance”. If this refers to VO2max in Table 1 it shows a significant decrease.
I find it hard to follow some parts of the manuscript,. Grammar checking is needed.
Author Response
Dear Editor
We greatly appreciate the interest taken in our manuscript and the time spent on its review. Thank you also for the opportunity to revise the manuscript. We have corrected the whole text and hope that it will meet the high standards of the Nutrients in the present form. Thank you in advance for reconsidering it for publication.
We highly appreciate the detailed valuable comments and constructive criticisms of the referees on our manuscript entitled: "8-day Fast and a Single Bout of Exercise: The Effect on Serum Methyl-Arginines and Amino Acids in Men." The suggestions are helpful, and we incorporate them in the revised paper. Please find responses to each inquiry or critique. Every comment was considered seriously and carefully.
Review 1
- The main issue of the paper is that I cannot assess the statistical analysis as it is not described in the text. Moreover, I found the discussion as an extension of the results while this section must be part of moving forward in the research field.
A: Thank you for this remark. A more detailed explanation of the statistics was included in a separate chapter.
- How does skeletal muscle metabolism respond to fasting? This would help to discuss your data and/ or introduce the research topic to some interesting data here (PMID: 36302454).
A: Thank you for suggestion, some changes has been introduced in the text and the reference has been introduced.
- Why was not measured NO? The authors extensively discuss how their results would influence NO biology.
A: Thank you for this remark. We appreciate your comment. We agree with the reviewer that data on NO could be a valuable addition to our results. However, the process of accurately and directly measuring NO is highly demanding. Given the technical challenges, we were unable to undertake its assessment. Nonetheless, we relied on existing studies that conclusively show an increase in ADMA, SADMA decreases NO synthesis.
- I think that the manuscript needs some language editing some parts are hard to follow.
A: We appreciate your observation. We have since engaged in a meticulous manuscript review to enhance its clarity and readability.
- Moderate comments:
- Reference for L45-47
A: Thank you for this remark. A suitable reference was added.
- References for L57-60
A: Thank you for this remark. A suitable reference was added.
- References for L65
A: Thank you for this remark. A suitable reference was added.
- Reference 6 must be further explained. Why inhibition of NOS is a pathological condition, explain.
A: We appreciate your comment. We have now provided a more comprehensive explanation. We hope that the additional details offer a sufficient understanding of the matter.
- Explain reference 7 in more detail. Why is it unclear?
A: Thank you for bringing this to our attention. We have since elaborated on this reference for greater clarity. We trust the added details will provide sufficient understanding.
- L 78- Peripheral or central fatigue?
A: Thank you for pointing out this ambiguity. To clarify, it refers to central fatigue. This information was added to the text.
- Was the blood collected immediately after exercise?
A: We appreciate your query. To clarify, the blood collection was performed 1-hour post-exercise. We have incorporated this information in the text for clearer understanding.
- “As for oxygen capacity indicating physical activity status, it decreased although without statistical significance”. If this refers to VO2max in Table 1 it shows a significant decrease.
A: We appreciate your keen observation. Indeed, we have since revised Table 1 to highlight the statistically significant findings and have also amended the associated explanation for consistency and accuracy.
Reviewer 2 Report
This eminent paper examines the biochemical effects of fasting on physical performance. I have some remarks:
- The authors investigate arginine-derived NO signaling modulators, but this list is much longer (e.g. NEMA and NOHA). How do the authors make sure there are no confounders in the given results?
- These by-products of arginine are chemical bases. Could the results have been influenced by the type of exercise (anaerobic vs aerobic)?
- Judging form the BIA results in Table 1 (especially at the start), the subjects were in a top physical condition. In contrast, fasting for an extended period of time is not recommended at all for people with medical conditions, and the authors should clearly state this in the paper.
English language requires little changes.
Author Response
Dear Editor
We greatly appreciate the interest taken in our manuscript and the time spent on its review. Thank you also for the opportunity to revise the manuscript. We have corrected the whole text and hope that it will meet the high standards of the Nutrients in the present form. Thank you in advance for reconsidering it for publication.
We highly appreciate the detailed valuable comments and constructive criticisms of the referees on our manuscript entitled: "8-day Fast and a Single Bout of Exercise: The Effect on Serum Methyl-Arginines and Amino Acids in Men." The suggestions are helpful, and we incorporate them in the revised paper. Please find responses to each inquiry or critique. Every comment was considered seriously and carefully.
Review 2
This eminent paper examines the biochemical effects of fasting on physical performance. I have some remarks:
- The authors investigate arginine-derived NO signaling modulators, but this list is much longer (e.g. NEMA and NOHA). How do the authors make sure there are no confounders in the given results?
A: We appreciate your point. Indeed, there could be other factors involved in the signaling process. However, we could not delve deeper into this aspect due to resource constraints. Nonetheless, we acknowledge the potential presence of these confounders.
- These by-products of arginine are chemical bases. Could the results have been influenced by the type of exercise (anaerobic vs aerobic)?
A: We appreciate your insightful query. Indeed, some data suggest that serum ADMA concentrations may vary between endurance and sprint-trained athletes. Furthermore, endurance exercise has been observed to increase ADMA levels, without affecting SDMA levels. However, in our study design and overall experiment plan, we implemented only one type of exercise and did not examine this potential differential effect. This is certainly an aspect worth exploring in future research.
- Judging form the BIA results in Table 1 (especially at the start), the subjects were in a top physical condition. In contrast, fasting for an extended period of time is not recommended at all for people with medical conditions, and the authors should clearly state this in the paper.
A: Thank you for this remark. Indeed, we emphasised it in the conclusions to provide clear guidance for readers.
Round 2
Reviewer 1 Report
The paper can be published as such.